# Mitigation of Climate Change for Urban Agriculture: Water Management of Culinary Herbs Grown in an Extensive Green Roof Environment

Stuart Alan Walters [1,*], Christina Gajewski [1], Amir Sadeghpour [2] and John W. Groninger [1]

1  School of Forestry and Horticulture, Southern Illinois University, Carbondale, IL 62901, USA
2  School of Agricultural Sciences, Southern Illinois University, Carbondale, IL 62901, USA
*  Correspondence: awalters@siu.edu

**Abstract:** Extensive green roofs provide space for local agriculture in dense urban environments. However, already extreme drought and heat conditions on green roofs are likely to worsen under future climates, challenging urban crop production and impeding food security. The potential productivity of annual and perennial culinary herbs on an extensive green roof (~8 cm depth) with minimal, but consistent, water inputs was evaluated within a humid, subtropical climate (Southern Illinois University-Carbondale, Carbondale, IL, USA). Vigor, growth, and overwintering ability of four different perennial culinary herbs, namely garlic chives (*Allium tuberosum*), lavender (*Lavandula angustifolia* 'Munstead Dwarf'), lemon balm (*Melissa officinalis*), and winter thyme (*Thymus vulgaris* 'Winter Thyme'), as well as vigor and growth of annual 'Italian large-leaf' basil (*Ocimum basilicum*) were evaluated under twice-weekly, weekly, and fortnightly water applications of 1 L to each plant. All species of perennial herbs produced greater dry perennial biomass and overwintering potential under the two most frequent water applications. Similarly, with weekly water applications, basil proved highly suitable for production in an extensive green roof environment. Weekly watering was required to provide commercially viable plant growth, vigor, and overwinter survival for all perennial herbs. These results indicate that supplemental water is an important consideration for sustaining culinary herb production on extensive green roofs with the increasingly hot and dry conditions provided under the climate change scenarios projected for cities currently experiencing temperate climates.

**Keywords:** drought stress; food security; irrigation; rooftop gardening; urban agriculture

## 1. Introduction

Climate change is a major impediment and challenge to food security throughout the world, with urban environments especially susceptible to food shortages. Anticipated climate change is likely to lead to more extreme weather events, which are oftentimes a source of disruptions to urban food supplies. Thus, urban agricultural activities become even more important in terms of helping to maintain food supplies to residents in highly populated areas. Since relatively little consideration has been given to the influence of projected climate change on urban agriculture, human populations in urban areas are at high risk of food insecurity from climate change [1]. Urban agriculture can make important contributions to food security in highly populated cities, which can in turn help build urban resilience and improve urban sustainability.

Alternative urban food production systems such as green roofs have gained popularity throughout the world in recent years [2]. However, extensive green roof systems were initially designed to aid in storm water management, with light-weight mediums developed to allow water flow through the medium's capacity; however, these systems only permit limited plant variation [3,4]. In contrast, intensive cropping systems emphasize growth mediums that balance drainage and aeration while providing consistent water supply to

the plant rooting environment. Water relationships between shallow soilless substrates in extensive green roof mediums and plants can have drastic effects on photosynthesis and evapotranspiration, which are important for establishing and maintaining crop plant productivity [5]. Healthy, actively growing plants are necessary to achieve adequate growth and development for crop production, but temperature extremes and drought conditions that result from shallow highly porous substrates oftentimes make this a challenge on green roofs [6]. Green roof mediums differ from natural soils since they contain an aggregate such as heat expanded clay as the base substrate [2,4]. Water holding capacity (WHC), permeability or water infiltration rate, density, pore volume and air-filled porosity, nutrient holding capacity, pH, and electrical conductivity (EC) are some of the substrate properties that affect plant performance [7].

Although plant selection is a key to successful plant growth on a green roof, water management is of the utmost importance for achieving and sustaining long-term productivity. The extreme environmental conditions typically found on rooftops impose challenges for the long-term survival of vegetation [8,9] through daily and seasonal temperature fluctuations [10,11]. Limited water availability, exposure to wind, solar radiation, and flash flooding can create harsh environments for plants growing in green roof environments [12]. Extensive green roof growth mediums, which are needed to support and anchor plants [13], tend to have shallow depths, are light-weight and well-drained, and are poor in organic matter and nutrients [14]. Therefore, the addition of nutrients and water will most times improve plant growth and establishment [15]. Drought is one of the most limiting factors in extensive green roof systems given shallow medium substrate depths (<15 cm) and reliance on natural precipitation to sustain growth [3]. Plant species vary in their transpiration rates and some plants require high amounts of water compared to others to grow and develop properly [16]. Thus, irrigation and water use are both important parts of green roof vegetable production systems when aiming to maintain proper levels of moisture for plant growth while preventing losses due to either over- or under-watering.

Water is an essential input for agricultural production, and water-efficient systems must be devised if urban agriculture is to play a meaningful role in contributing to food security. However, due to ongoing population growth, urbanization, and climate change, competition for water resources among essential urban uses, including for direct human consumption, can be expected to increase. Thus, crop water-use efficiency becomes a crucial part of agriculture production systems and improving water-use efficiency of crops is now a primary focus of agriculture regarding the improvement of food security. Moreover, climate change predictions show clear increases in temperatures and a concomitant increase in potential evapotranspiration, which lead to more frequent episodes of climatic anomalies such as droughts and heat waves [17]. Culinary herbs are popular plants to grow in green roof environments due to demand from local restaurants and consumers seeking a fresh local supply for use in food preparation. Dietary trends in recent decades have shown the increasing popularity of fresh culinary herbs [18]. However, despite increasing in popularity, there is limited information regarding the culinary herb irrigation management practices needed to maintain productive plants [18]. To address this knowledge gap, an irrigation evaluation using minimal water inputs was conducted on an extensive green roof to determine the drought tolerance of basil (*Ocimum basilicum*) and several perennial herbs, including garlic chives (*Allium tuberosum*), winter thyme (*Thymus vulgaris*), lavender (*Lavandula angustifolia*), and lemon balm (*Melissa officinalis*). Additionally, the influence of soil moisture applications on the overwintering ability of the perennial culinary herbs will also be determined.

## 2. Materials and Methods

This study was conducted in the summer of 2018 and 2019 to evaluate various water application treatments on the growth and productivity of four perennial culinary herbs (garlic chives, winter thyme, lavender, and lemon balm) and an annual herb ('Italian large-leaf' basil) on the Southern Illinois University-Carbondale (SIUC) campus green

roof (37.712993° N Latitude, 89.22227° W Longitude), which has a humid subtropical climate [19]. The extensive green roof on the SIUC Agriculture Building represented the state-of-the-industry at the time of its construction in 2010 and consists of several roof layers; a waterproof membrane, a protective fabric layer, a root permeable layer, a drainage layer, a filter fabric layer, and a growing medium constrained by an edging system [20]. The drainage layer is a plastic egg-crate-type tray system that collects rainwater and allows excess water to drain from the roof. The extensive green roof medium was obtained from the Midwest Trading Company (Virgil, IL, USA) and is predominately a mineral based expanded light-weight clay aggregate containing ~4 to 5% organic matter at establishment. During installation in 2010, the green roof medium also included 0.91 kg/m$^3$ of slow-release nitrogen fertilizer and 3.63 kg/m$^3$ of iron sulfate. However, significant amounts of fertility and organic matter (vermicompost) have been added and extracted through research plot use on the SIUC green roof over the years. This area in which herbs were grown has supported past research on tomatoes, lettuce, and melons. Recent soil analyses have indicated that the medium now has a pH of 7 to 7.5, organic matter content of 4 to 5%, and a C/N ratio of 14 to 17. This green roof can support 11 kg/m$^2$, which limits the depth of the soil-less medium to approximately 8 cm [20]. The average temperature for the green roof environment during the summer months ranges from greater than 35 °C (day) to about 20 °C (night). Temperatures are highly variable during the winter months with highs reaching up to 15 to 20 °C on certain days and lows sometimes below −15 °C, while daytime and nighttime average temperatures range from 0 to 10 °C and 0 to −7 °C, respectively. Most perennial herbs evaluated in this study would not be significantly damaged until temperatures reach below −7 °C. Annual local precipitation totals were 1579 mm and 1472 mm for 2018 and 2019, respectively, while growing season totals (May–October) were 768 and 742 mm for those same years, respectively.

### 2.1. Perennial Herb Experiment

2.1.1. Experimental Setup

The perennial herb experiment was set up as a 3 × 4 factorial treatment arrangement in a randomized complete block design (RCBD) to evaluate three watering regimes (once per week, twice per week, or once every two weeks), with 1 L of water representing approximately 13 mm rainfall, which was applied by gently pouring the water onto the medium around each plant per application. Thus, the application of 1 or 2 L of water weekly would provide the equivalent of 13 and 25 mm of rainfall per week, respectively, while 1 L every two weeks would equal 6 mm of weekly rainfall. The total amount of water applied to each plant on a per month basis was 2, 4, and 8 L for the once per week, twice per week, or once every two-week treatments, respectively. All watering regimes were supplemental to natural precipitation. The perennial herbs were chosen for their popularity in regional markets and ability to survive hot, dry summers. All perennial herb transplants were started in a greenhouse and placed in a cold frame to harden off at the 3- to 4-leaf stage for 2 weeks prior to planting. Herb plants were at the 4 to 6 leaf-stage at transplanting. Four plants of each perennial herb were planted in mid-May each year in each experimental unit and spaced 15 cm within and between rows, with water application treatments started soon thereafter. In June, ~5 g of Osmocote® slow-release fertilizer (14N-14P-14K; The Scotts Company, Marysville, OH, USA) was applied around each plant, with a second application of the same amount applied in late August.

2.1.2. Data Collection and Analysis

Measurements of the perennial herb stem diameter, plant height, leaf number, foliar wilt rating, and fresh and dry weight of each plant were collected in late October each year. Foliar wilting status was evaluated on a scale of 1 to 3 = low, 4 to 6 = moderate, and 7 to 9 = high amounts of visible unwanted foliar wilt. Overwinter survival of all perennial herb plants was assessed the following year after planting, which was the last week of April in 2019 and 2020. All data were subjected to analysis of variance procedures appropriate for

a factorial experimental design using the GLM procedure of SAS (SAS Institute, ver. 9.4, Cary, NC, USA). Fisher's protected least significant difference (LSD) test at $p \leq 0.05$ was generated by SAS and used to separate treatment means.

*2.2. Basil Experiment*

2.2.1. Experimental Setup

The basil experiment evaluated the same three watering regimes (once per week, twice per week, or once every two weeks; 1 L of water applied to each plant) in terms of the growth and productivity of this annual herb. The experiment used a RCBD design with three replications, with four basil plants grown in each experimental unit. Basil transplants were started in a greenhouse and transplanted to the green roof in mid-May each year, with watering regimes commencing soon thereafter. Basil plants were spaced approximately 0.5 m within and between rows. In June each year, an application of ~5 g of Osmocote® slow-release fertilizer was placed around each plant, with a second application of the same amount applied in late August.

2.2.2. Data Collection and Analysis

Basil plant heights and diameters were collected at harvest in October 2018 and 2019. The total yield (kg) was determined for each plant harvested. Basil plant leaf number, stem diameter, height, and weight, along with total fresh and dry plant weight, were collected at the end of the season. All data were subjected to analysis of variance procedures using the GLM procedure of SAS (SAS Institute, ver.9.4, Cary, NC, USA). Fisher's protected least significant difference (LSD) test at $p \leq 0.05$ within SAS was used to separate treatment means.

**3. Results**

*3.1. Perennial Herb Experiment*

3.1.1. Data Analysis

The data analysis indicated no cultivar or watering regime by year interactions ($p > 0.05$) for the dependent variables evaluated; thus, data are shown combined over years (Table 1). There was also no cultivar by water interaction observed for the plant parameters collected ($p > 0.05$); thus, data for all four perennial herbs were combined to show the effect of water treatments on foliar wilt, stem height, and fresh and dry weights. For overwintering ability, no interaction of year with perennial herb or irrigation timing was detected ($p > 0.05$); thus, data are shown combined over both years. However, an interaction between perennial herb and watering regime was detected for winter survival rates ($p \leq 0.05$); therefore, irrigation timing is shown for each perennial herb (Table 2).

**Table 1.** Influence of irrigation regime on the foliar wilt of perennial herbs and the resulting growth characteristics in an extensive green roof environment.

| Water (L) Applied Each Month per Plant | Foliar Wilt Rating | Stem Height (cm) | Total Plant Fresh wt. (g) | Total Plant Dry wt. (g) |
|---|---|---|---|---|
| 2 (once per two weeks) | 6.2 c | 16.6 c | 0.068 a | 0.047 b |
| 4 (once per week) | 3.3 b | 21.4 b | 0.071 a | 0.052 a |
| 8 (twice per week) | 2.6 a | 24.8 a | 0.070 a | 0.052 a |

Data are means of three replications. At each specific application timing, 1 L of water was applied to each plant in an experimental unit. The perennial herbs evaluated were garlic chives (*Allium tuberosum*), lavender (*Lavandula angustifolia* 'Munstead Dwarf'), lemon balm (*Melissa officinalis*), and winter thyme (*Thymus vulgaris* 'Winter Thyme'). Foliar wilt rating: 1 to 3 = low, 4 to 6 = moderate, and 7 to 9 = high amounts. Means within a column followed by the same letter do not differ significantly according to Fisher's protected LSD, $p > 0.05$.

**Table 2.** Perennial herb winter survival percentage as influenced by irrigation application timing and species in an extensive green roof environment, combined over the 2018 and 2019 growing seasons.

| Water (L) Applied Each Month per Plant | Garlic Chives | Lavender | Lemon Balm | Thyme |
|---|---|---|---|---|
| 2 (once per two weeks) | 80.1 b | 55.0 b | 91.7 b | 87.5 a |
| 4 (once per week) | 87.5 a | 95.8 a | 95.8 a | 87.5 a |
| 8 (twice per week) | 91.7 a | 95.8 a | 95.8 a | 91.7 a |

Data are means of three replications. At each specific application timing, 1 L of water was applied to each plant in an experimental unit. The perennial herbs evaluated were garlic chives (*Allium tuberosum*), lavender (*Lavandula angustifolia* 'Munstead Dwarf'), lemon balm (*Melissa officinalis*), and winter thyme (*Thymus vulgaris* 'Winter Thyme'). Means within a column followed by the same letter do not differ significantly according to Fisher's protected LSD, $p > 0.05$.

### 3.1.2. Influence of Irrigation Frequency on Foliar Wilt and Stem Heights

Foliar wilt and resulting stem heights differed ($p \leq 0.05$) among irrigation treatments for all perennial herbs evaluated (Table 1). Higher amounts of foliar wilt were observed with the application of 1 L of water every two weeks compared to either once or twice per week, both of which had similar low amounts of foliar wilt (Figure 1). Those plants receiving only one water application every two weeks had a foliar wilt rating of 6.2, which was about a 1.9 and 2.4 x increase in visual foliar wilt compared to applying water either once or twice per week, respectively. The greatest stem heights were observed when 1 L of water was applied twice per week, followed by the once per week application, with the once every two weeks application providing the least. The perennial herbs had 29% and 49% greater stem height for the once and twice per week water application, respectively, compared to applying water once every two weeks.

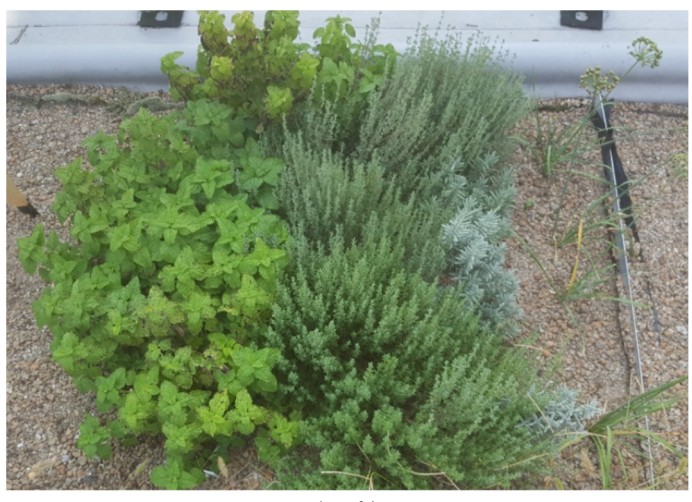 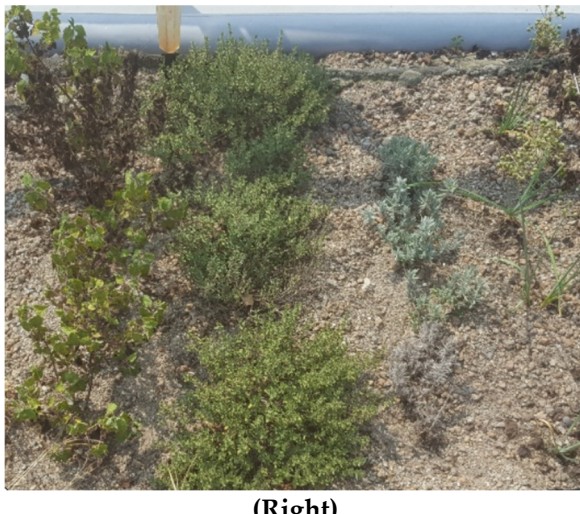

**(Left)** **(Right)**

**Figure 1.** Comparison of watering regimes (1 L per plant) of twice per week (**left**) and once every two weeks (**right**) for the green roof perennial herb experiment.

### 3.1.3. Influence of Irrigation Frequency on Biomass

Although perennial herb fresh weights obtained at the end of the growing season were not affected by irrigation regime ($p > 0.05$), dry weights did differ (Table 1). Water application to plants either once or twice per week provided greater dry perennial herb biomass than applying water once every 2 weeks; the dry weight growth increase was ~10% for those perennial herb plants that received weekly water applications. Regardless of the perennial herb evaluated, some weekly watering was required to maintain adequate plant growth and vigor in an extensive green roof environment.

### 3.1.4. Influence of Irrigation Frequency on Winter Survival

Irrigation regime also affected perennial herb winter survival rates (or overwintering ability). The once and twice per week applications of 1 L of water per plant improved the overwintering survival rates of garlic chives, lavender, and lemon balm. The winter survival rate of garlic chives was improved by 9 and 15% for the once and twice per week water applications compared to only watering once every two weeks. Additionally, the application of water at weekly intervals was critical to the successful overwintering of lavender, and a 74% decrease in overwintering survival rates was observed when watering only once every two weeks compared to the weekly applications. However, the overwintering of lemon balm slightly improved (5%) when water was applied either once or twice weekly compared to once every two weeks. In comparison, thyme appeared to be more drought tolerant than the other perennial herbs species evaluated, since winter survival rates did not differ among water application treatments ($p > 0.05$) for this perennial herb. Although the twice per week water application slightly improved the overwintering survival rate of thyme by about 5% compared to the other less frequent water applications, this treatment did not differ ($p > 0.05$) from the less frequent water application treatments.

### 3.1.5. Overall Irrigation Frequency Influences

Infrequent watering of perennial herbs in an extensive green roof environment located in a humid, subtropical climate having hot summers with sporadic and infrequent rainfall events will most likely result in lower plant growth, vigor, and productivity and reduce the overwintering potential of some perennial herbs. Thus, weekly water applications during the growing season are necessary to sustain perennial herbs from year to year in this environment. Less frequent watering will not only reduce perennial herb plant growth and productivity in an extensive green roof environment but also lead to plant loss from one growing season to the next.

### *3.2. Basil Experiment*

### 3.2.1. Data Analysis

Basil foliar wilt results indicated that there was a difference ($p \leq 0.05$) among watering regimes (Table 3). The least amount of wilt occurred at the irrigation application timing of 1 L of water applied to a basil plant twice per week, followed by the once per week application, with most occurring in the once every two weeks application (Figure 2). The foliar wilt rating was highly correlated to basil fresh (r = −0.74; $p < 0.0001$) and dry plant biomass (r = −0.59; $p < 0.0001$), stem number (r = −0.79; $p < 0.0001$), diameter (r = −0.61; $p < 0.0001$), height (r = −0.88; $p < 0.0001$), and leaf number (r = −0.93; $p < 0.0001$). This indicates that the amount of unwanted foliar wilt observed based on the watering regime was an effective indicator of basil plant growth by the end of the growing season.

**Table 3.** Influence of watering regime on basil growth in an extensive green roof environment.

| Water (L) Applied Each Month per Plant | Foliar Wilt | Plant Leaf No. | Stem No. | Stem Diameter (cm) | Plant Height (m) |
|---|---|---|---|---|---|
| 2 (once per two weeks) | 2.5 c | 268 c | 13.3 c | 1.0 b | 0.22 c |
| 4 (once per week) | 5.0 b | 496 b | 15.8 b | 1.3 a | 0.28 b |
| 8 (twice per week) | 7.0 a | 683 a | 19.2 a | 1.4 a | 0.37 a |

Data are means of three replications. At each specific application timing, 1 L of water was applied to each plant in an experimental unit. Foliar wilt was rated on a scale of 1 to 3 = low, 4 to 6 = moderate, and 7 to 9 = high amounts. Means within a column followed by the same letter do not differ significantly according to Fisher's protected LSD, $p > 0.05$.

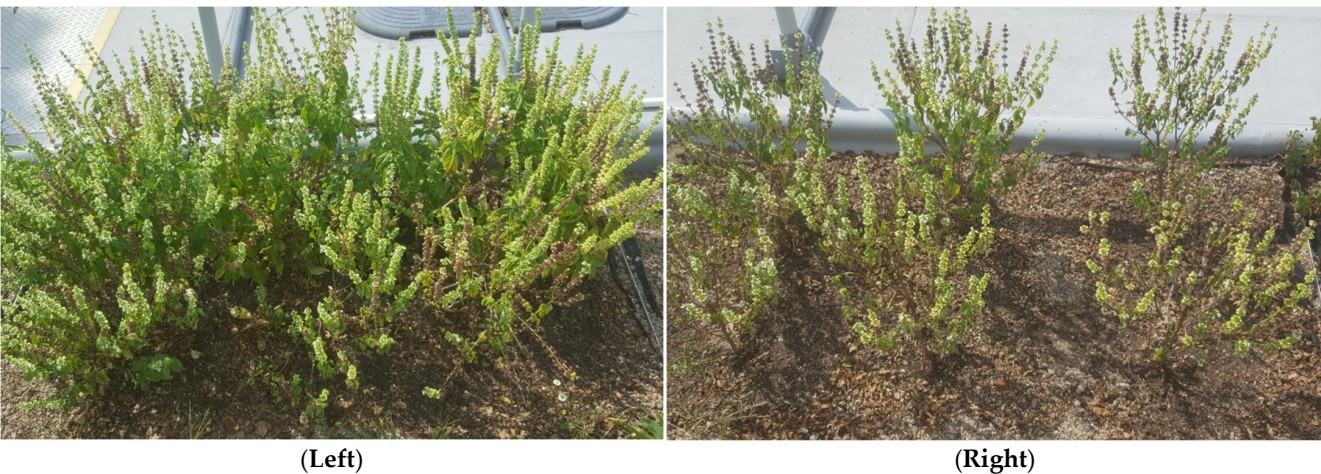

| (**Left**) | (**Right**) |

**Figure 2.** Comparison of watering regime treatments showing the effects of 1 L of water applied twice per week (**left**) and once every 2 weeks (**right**) on resulting basil foliar growth in an extensive green roof environment.

### 3.2.2. Influence of Irrigation Frequency on Growth Characteristics

Basil plant growth characteristics were affected by watering regime (Table 3). Plant leaf number was 85% and 155% greater for once and twice per week application timings, respectively, compared to applying water only once every two weeks. Similarly, stem numbers, diameters, and heights followed similar trends. Basil stem numbers increased by 19% and 44% for the once and twice per week irrigation application timings, respectively, compared to application timings of once every two weeks. Basil stem diameters were improved 30% and 40%, respectively, for the irrigation application timings of once and twice per week compared to the once every two weeks application timing, although there was no difference ($p > 0.05$) between the once or twice per week 1 L water application to basil plants. Lastly, basil plant heights increased 27% and 68% for the irrigation timings of once and twice per week, respectively, compared to the once every two weeks application timing. The 1 L application of water once or twice per week to each basil plant provided the largest increase in basil leaf number, stem number, stem diameter, and stem height. Basil plant fresh and dry weights also differed ($p \leq 0.05$) among the irrigation timings evaluated (Table 4). Water applied twice a week provided greater basil fresh and dry weights, followed by the once-a-week water treatment, with the once every two-week application providing the least. For basil fresh weight, 21% and 36% increases were observed when water was applied twice a week compared to the once per week or every two weeks application, respectively. Additionally, a 26% and 36% basil dry weight increase was observed in the twice per week irrigation application compared to water applied once per week or every two weeks, respectively.

**Table 4.** Influence of irrigation regime on basil fresh and dry weights in an extensive green roof environment.

| Water Applied Each Month per Plant (L) | Plant Fresh wt. (g) | Plant Dry wt. (g) |
| --- | --- | --- |
| 2 (once per two weeks) | 0.075 c | 0.047 b |
| 4 (twice per week) | 0.091 b | 0.059 a |
| 8 (twice per week) | 0.098 a | 0.064 a |

Data are means of three replications. At each specific application timing, 1 L of water was applied to each plant in an experimental unit. Means within a column followed by the same letter do not differ significantly according to Fisher's protected LSD, $p > 0.05$.

### 3.2.3. Overall Irrigation Frequency Results

These results indicate that basil requires significant amounts of moisture to grow and develop properly in an extensive green roof environment. Moreover, this experiment suggests that a limited water supply will sustain basil plants in an extensive green roof environment. However, to maximize productivity, greater than weekly water applications are required in this typically stressful, hot, and drought-prone environment.

## 4. Discussion

These results highlight challenges but also confirm the viability of growing herbs in an extensive green roof environment. Although green roofs provide a harsh and stressful growing environment for plants [3,12] they are recognized as becoming an increasingly important part of urban agriculture and are key to making cities more sustainable and habitable [21–23]. Although the use of extensive green roofs could contribute a significant portion of the food used by urban populations, water management and irrigation system efficiency are major challenges when considering growing edibles on green roofs [2]. Our results demonstrate that, with a consistent supply of water, marketable food crops can be produced on an extensive green roof.

As climate change strains water resources, both by limiting supply and increasing evapotransporative demand on crops, urban cropping systems must take both water use efficiency and crop resiliency into account. Lack of substrate moisture is one of the most limiting factors for extensive green roof systems given shallow medium substrate depths (<15 cm). In our study site's subtropical humid climate, natural precipitation alone can sustain growth of only the hardiest, and non-edible, ground covers [3,12]. Therefore, supplemental irrigation is an essential component of edible crop production systems for extensive green roofs, since adequate and consistent moisture is necessary to maximize produce quality and yield [2,24]. Irrigation methods using small amounts of water to produce edible crops are most important in urban green roof environments in terms of sustaining productivity over time. Sophisticated irrigation systems using commercially available sensors that register soil moisture can be used, but they may not work properly with green roof mediums, which are often too porous for proper functioning [4]. Moreover, the rate of evaporation or utilization of the reserved water depends on numerous factors, such as temperature, wind conditions, and the types of plant materials used; during periods of no rainfall, supplemental irrigation is critical when providing water to plantings, which is especially crucial when integrating food production onto characteristically stressful green roof environments [2].

Extensive green roofs have the potential to supply high demand food crops to local consumers, but there are still challenges that need to be overcome to maximize their productivity in these settings [2,24,25], especially under changing climatic conditions. Our study indicated that annual and perennial herbs grown in the extreme environmental conditions found on extensive green roofs will still have some productivity, even with minimal water applications. The growth of lemon balm, chives, thyme, and lavender was sustained with at least 1 L of water applied per plant every other week for these herbs. However, as more water was applied, the productivity and overwintering potentials of these perennial herbs were improved. Lavender was most negatively affected by the lack of water compared to the other herbs evaluated. Although less frequent water applications provided lower winter survival rates for nearly all perennial herbs evaluated (except for winter thyme), lavender was most affected. A 74% decrease in lavender overwintering survival rates was observed when plants received only 1 L of water once every two weeks compared to weekly irrigation applications. Moreover, both lemon balm and thyme had high overwintering rates each year, even with minimal water applications, and had very prolific growth and spreading habits. Additionally, basil (an annual herb) was very responsive to water applications in an extensive green roof environment, with the highest plant growth characteristics observed when 1 L of water was applied per plant twice a week compared to the other irrigation regimes evaluated.

Our study supports the critical need for irrigation in an extensive green roof environment when growing annual and perennial herbs, with supplemental irrigation helping to obtain growth and vigor adequate for food production and improvements in the overwintering survival of perennial herbs. Drought stress in this environment will contribute toward herb plants having less growth and biomass by the end of the growing season. However, water deficits improve quality for several leafy crop species through increases in essential oils, aroma, and plant metabolites that contribute to overall antioxidant potential [18]. Under hot and dry conditions, aromatic plants produce larger quantities of essential oils, which are secondary metabolites found in single or multicellular glands located on the epidermis of leaves and flowers or trichomes [26]. The essential oil properties of Mediterranean herbs (e.g., basil, lavender, oregano, or thyme) under reduced moisture conditions can easily be improved in both intensive and extensive green roof systems. Although overall herb productivity may be decreased with less moisture, the essential oils and antioxidant potential may more than compensate for yield loss when grown in these dry environments, depending on their final use.

This study indicates that both annual and perennial herbs can be grown effectively on extensive green roof environments, although at least weekly water applications were required during the growing season to ensure adequate growth and biomass accumulation. Due to the low depth and lack of organic matter in extensive green roof mediums, providing sufficient moisture holding capacities can be a challenge when seeking to achieve maximum crop growth. However, it is noteworthy that our study site, in the lower midwestern United States, has a generally hotter and drier growing season compared to most green roofs in North America and Western Europe, which may be a useful predictor for conditions and productivity potentials for locations under future climate change scenarios (19).

The watering rates used in this study were based on a minimum application rate to ensure crop plant survival. Further refinement of this system will require situation-specific application rates. Green roof-specific sensor system technologies adapted to rapidly changing plant and atmospheric demand for water would increase opportunities to manage extensive green roofs for food production. Improvements in roof media, water retaining gels, mulching, and subsurface irrigation systems would similarly increase opportunities to manage extensive green roofs for food production [2,27]. Further research is needed to also address issues that can affect crop quality under the stressful conditions found in green roof environments (26). Sedum (*Sedum album*) groundcover can possibly be used as a substitute for watering hardy culinary herbs on green roofs [28] due to high amounts of medium coverage and reduced water losses occurring with these plants. Additionally, those culinary herbs (and varieties) most suited to green roof production should be determined in a more extensive evaluation, as this would provide information on their drought tolerance and identify which would grow best in the hot, dry environments that will most likely be exacerbated during climate change. Additional investigations could also evaluate the tradeoffs associated with maintaining herb crops compared to other alternative vegetation covers and possibly allow extensive green roofs to better fulfil cities' goals to balance food security, water conservation, and climate mitigation needs.

## 5. Conclusions

Although culinary herbs were successfully produced on a heat- and drought-prone extensive green roof, crop plant growth, vigor, and overwintering performances were positively correlated with the amount of supplemental water provided. Although rooftops can be used to create spaces to produce edible crops, water management must be considered as an important factor when maximizing the productivity and overwintering potential of perennial herbs. These results highlight the potential of green roof urban agriculture to improve food security, economic opportunities, and community development in cities that face ecological challenges due to ongoing climate changes. However, drought is one of the most limiting factors in extensive green roof systems given the shallow medium substrate depths (<15 cm) and the reliance on natural precipitation to sustain plant growth.

Anticipated climate change can be expected to exacerbate this stress, especially through intense heatwaves and extended drought periods that threaten water availability and diminish herb plant growth in these already stressful green roof environments. Regardless, supplemental irrigation is an essential component of culinary herb crop production systems for extensive green roofs, since adequate and consistent moisture is necessary to maximize quality and yield.

**Author Contributions:** Conceptualization and methodology, S.A.W. and C.G.; formal analysis, C.G. and S.A.W.; investigation, C.G.; resources, S.A.W.; data collection, C.G.; writing—original draft preparation, C.G. and S.A.W.; writing—review and editing, S.A.W., A.S. and J.W.G.; supervision, S.A.W., A.S. and J.W.G. All authors have read and agreed to the published version of the manuscript.

**Funding:** This research received no external funding.

**Data Availability Statement:** Not applicable.

**Acknowledgments:** University of Illinois Extension provided personnel support for C.G. to complete this project as an MS thesis, while Walters' research program provided all other monies and commodities used this project.

**Conflicts of Interest:** The authors declare no conflict of interest.

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
