# Peer review of "Mitigation of Climate Change for Urban Agriculture: Water Management of Culinary Herbs Grown in an Extensive Green Roof Environment"

_climate, doi:10.3390/cli10110180_

Round 1

Reviewer 1 Report

General Comments

This is an interesting research manuscript on water management of culinary herbs grown in green roof environment.

·       The title of the manuscript relates the research with the issue of climate change. Indeed, the discussion on climate change has been extensively incorporated in the introduction and conclusion of the research. However, the authors should also have emphasized, how the employed research methodology was design to simulate the climate change impact (drought, heatwave, etc.) on the growth of culinary herbs grown on green roof environment. Otherwise, the research seems to only reflect the irrigation frequency needs of the culinary herbs.

·       Why did you water the plant once per week, twice per week…does this reflect any drought condition due to climate change? Please provide the intention for this watering frequency in the face of climate change. Otherwise, this is just to measure the watering frequency of the culinary herbs.

·       The research output also lacks the discussion on it reflection on the climate change impact. For example, does the low growth of culinary herbs with watering frequency of once in two weeks, really due to climate change impact? Or is it just reflected the watering frequency needs for the plant?        

·       Considering other anthropogenic nature of green roof environment, that mostly influence by human changes, technology, and mechanical aspect of growing culinary herb, is there any real event, that can be used to validate, that indeed climate change has cause a significant impact on green roof environment?

Although the research manuscript has been well design with robust methodology and good result, the authors seems to fail in incorporating how the methods and result really reflect the impact of climate change. Therefore, I hope the authors can improve this aspect of the research manuscript to really reflect how this research can “mitigate” the impact of climate change as intended in the research title. At this juncture, please consider major revision of the manuscript. The specific comments can be found below.

Specific comments

2. Materials and Methods

·       Line 127: caontent > content

2.1.2. Data collection and analysis

·       Line 160: follwing > following

Author Response

We have taken your suggestions in the new version of the manuscript. Please see our comments.

Reviewer 2 Report

Provided climate change is limiting water availability to such an extent as you argued in the introduction, how can you support this completely water-dependent urban green roof 'agriculture', alone for restaurants or 'family use'? - Please try to reduce redundancy, add and correct some methodical details, check my comments given in the annotated PDF attached.

Author Response

We have taken your suggestions and incorporated many into the new version of our manuscript.  Please see our comments. 

Reviewer 3 Report

Very nice research. My comments:

-          the watering or irrigation system is crucial for the plant grown, no debate about this, but why did you don´t try to use some kind of easy use watering system ,e.g. drip irrigation? Because every day irrigation with this type and totally less water will be maybe more efficient. Are you trying to test it in the future?

-          what is the water retention capacity in your used roof? I missed it in the paper, also the thicknesses of the roof composition.

-          what was the rainfall in the different years? Did it differs or creates difference? I am not aware of the rainfall in Illinois, but it maybe doubled the irrigation amount.

-          the watering with the specific water amount was applied on the plant leaves like rain or directly to the substrate surface? When applying one liter in short time on the soil, most of the water will probably flow through the composition through the or in the retention layer. Were the roots of the plants able to recover the water later?    

-          once per week, twice per week, or once every two weeks; with 1 L of water applied to each plant at each application timing). The 1 L of water application represented approximately 13 mm of rainfall. Thus, applying 1 or 2 L of water weekly would provide about 13 and 25 mm of rainfall per week, respectively, while 1 L every two weeks would equal 6 mm of weekly rainfall. These calculations are IMHO wrong, because the intensity and the cumulative rainfall differs significantly from pouring the water on one plant.

Fisher’s protected LSD citation will be appreciated

l. 122the green roof medium also included 1 kg/m slow-release nitrogen fertilizer and 4 kg/m of iron sulfate   - are the amounts per square meters or thickness of the substrate?

l 127 caontent

Table 1 – meaning of a, b, c in the columns is not properly described

Figure 1 – better readability will be achieved by gap between the figures and probably better is to describe the left figure first.

Table 3 - stem diameter and stem height – these numbers seem strange, probably are wrong with the unit.

Rowland et al. (2018) found strong evidence that water deficits will improve crop quality of several leafy herb crops, which included increases in essential oils, aroma and quality  - this needs to highlighted also in the conclusion – because it is in contrary to your conclusion. So it depends if your plants are for foliar/ leaves or for other things.

The green roofs benefits are mostly achieved with the surface totally covered with some kind of plants. This is not possible with the herbs planted in big distances or too sparsely.  Maybe this experiment should be compared together with temperature measurements.

Author Response

We have taken your comments into consideration in the new version of the manuscript.  Please see our comments.

Reviewer 4 Report

This article is dedicated to demonstrate the relationship between urban green roofs and urban agriculture. For this object, Four different species of culinary herbs were selected to derive indices related to the cold resistance and overwintering ability of culinary herbs for urban agriculture, through practical experiments, and key coefficients such as growth and productivity of the study subjects were scientifically evaluated. And several important and interesting results and key factors that should be taken into account when combining green roofs with urban agriculture were summarized.

However, the manuscript contains some Subtle deficiencies that need to be corrected before the publication.

1)      Firstly, in the introduction section, the notation of previous studies does not follow the order of references section, it is recommended to follow the order of references.

2)      There is not enough summarization of previous studies in introduction, which supporting the theoretical basis of the article, therefore it is recommended to Improve related contents.

3)      In method and materials section, there is no explanation about why these four subjects were chosen for the study, in order to increase the credibility of the article results it is recommended to add this content.

4)      To show the overall structure of this paper more vividly, it is recommended to add a technology roadmap in method and material section.

5)      In the method section, should cite previous studies to demonstrate the reliability of each experiment applied in this article.

6)      The discussion section should focus on verifying the reliability of each experimental results.

7)      The discussion section lacks the shortcomings and future prospects of this study.

8)      It is recommended that the discussion section be divided into smaller chapters for validation of results, future prospects, etc.

9)      In the section of conclusion, it is supposed to be a brief description about the findings of this study only. Therefore, the conclusion section needs to simplify.

10)  For a more standardized structure of the entire text, it is recommended that all tables be converted to three-line tables.

Author Response

We have taken your comments into consideration when rewriting the new version of the manuscript. Please see our comments.

Round 2

Reviewer 1 Report

The authors justify the incorporation of climate change in the study to other area that might face a similar future climate condition. I believe the right term for it is, future 'climate similarity' of other region in the world with the study area. I still did not convince that sufficient climate change (future climate similarity) discussion has been incorporated throughout the research. The research is nonetheless important for agriculturist in urban area, and the output is important for the climate action in urban environment. I will leave it to the editor judgement to accept the paper.

Reviewer 2 Report

You succeeded in improving your presentation. From my side I'll propose to publish it now. Especially I found your response file well fitting to my own style, thank you.

Author Response

Reviewer 2 stated after the first review...You succeeded in improving your presentation. From my side I'll propose to publish it now. Especially I found your response file well fitting to my own style, thank you.  So no changes in this round were made.

Reviewer 3 Report

The paper is improved, but in my oppinion more of my crucial comments are not considered or asnwered properly. Please, try to respond to it. Thank you.

Round 3

Reviewer 3 Report

Thank you very much for the answers. I hope , that my comments increased the quality of the paper.

I have still problem with the table 3. From the manuscript number 1, stem height was changed from cm to meters. According, that you are used to measured the lengh with feets, are you sure that the stem height is between 3 - 4 feets?

The same is for the diamter of the stem. 2,5 cm diameter is like an inch. Are you sure?

And regarding the gap between the figures in Figure 1. In Fugre 2, there is gap between the pictures since the first version of the manuscript.

Thank you.

Author Response

Thanks for your thorough review and I appreciate all that you have done to make this a better manuscript. 

You are correct regarding Table 3, with so many authors involved in the review and editing process, I just missed that.  Those were actually feet for the measured plant heights, and inches for the stem diameters as you pointed out. We had corrected the headings by not the data below. That is now corrected. Good catch!  

I also removed the space or gap between the photos in Figure 2.

Thanks greatly for your input.  Dr. Alan Walters